# Exploring Citizens Perception of the Police Role and Function in a Post-Colonial Nation

Michael Mathura

School of Justice Studies, Liverpool Centre for Advanced Policing Studies, Liverpool John Moores University, Liverpool L3 5UZ, UK; m.mathura@ljmu.ac.uk

**Abstract:** Before attempting to develop productive and harmonious working relationships between citizens and the police in a post-colonial society such as Trinidad and Tobago (T&T), it is imperative to initially gain a more precise understanding of the role and function of the police. This qualitative study suggested that the current role and function of officers is parallel to the colonial model of policing, where officers operated in a paramilitary manner. This model of policing was concerned with law enforcement and public order duties, which was highlighted as counterproductive for police and public relations. The model was also popular for police treatment based on citizens socioeconomic status. The results of this study suggest that police officers should implement a Service Oriented Policing approach (SOP), which could allow police officers to become proactively involved with communities and citizens, build stronger and increasingly productive relationships and be more effective and efficient as an institution.

**Keywords:** Trinidad and Tobago; policing; role and function; service oriented policing

## 1. Introduction

### 1.1. Overview

The police institution has been in existence for an exceptionally long time and its role and function often becomes debatable (Manning 2014; Bowling et al. 2019). On one hand, the public need the police for protection; on the other, citizens often criticised the police for what they do and how those actions are performed (King 2009; Mathura 2019). In a post-colonial society such as Trinidad and Tobago (T&T), this is consistent.

As human beings, people often have different views about the police, such as, loud sirens, flashing blue lights, high speed chases, officers in blue uniform with truncheons and officers speaking with school children about safety and the law (Uglow 1988). Most occupations are affiliated to a specific activity, for example, teachers provide education, doctors provide health care and electricians provide electricity. However, the police are not limited to law enforcement specifically (Uglow 1988; Reiner 2010). Therefore, it is often difficult to align the role and function of the police to a specific paradigm. However, regardless of the task, their roles eventually become integrated into communities and the lives of citizens (Uglow 1988; Reiner 2010).

Since policing is interconnected with the lives of citizens, working relationships with trust and confidence is vital (Boateng 2012; Farrow 2022). When citizens trust police, it is more likely that they [people] would become interested in policing initiatives and develop voluntarily participation. However, when this is not possible, community decay, loss of lives, lack of communication and support for police initiatives could develop (Worden and McLean 2017; Farrow 2022). In T&T, the police are responsible for maintaining peace and order, minimising fear and victimisation of crime, and they are the most visible symbols of state power, authority, and law enforcement amongst other agencies (Wallace 2011; Maguire et al. 2017).



### 1.2. Statement of Problem

The nascent body of research on policing in T&T have suggested that there are several problems obstructing the country's police service from becoming modern, effective, efficient, and service-oriented. According to Pino and Johnson (2011) many police officers in T&T have been accused of profiling citizens, coercive treatment, and corruption. Citizens were fearful of the police, and this contributed to poor trust and confidence. Some police officers were also accused of being involved with criminal gangs which led to legitimacy concerns (Townsend 2009; Her Majesty's Inspectorate of Constabulary and Fire and Rescue Services 2019).

According to King (2009), the Trinidad and Tobago Police Service (TTPS) had minimal changes from its colonial heritage and the institution continued to operate in a paramilitary manner which was counterproductive for a harmonious relationship between police and citizens. An important aspect of paramilitary policing is armed officers who relied on coercive tactics to resolve problems in the communities. Samad (2011) suggested that due to the brutal force and negligence of some police officers in T&T, there were several incidents of fatalities and human rights concerns.

Police officers in T&T are often accused of profiling and victimising citizens from disadvantaged communities, especially during stop and search operations (Townsend 2009). It was suggested that the suppression of the weaker class often fostered a divide between the police and citizens leading to communication decay. Therefore, officers were unable to resolve community problems and crime became widespread (Deosaran 2002; Wallace 2011). The Committee on the Restructuring of the Police Service (1984) described the Trinidad and Tobago Police Service (TTPS) as:

> "*A repressive force ready to harass people at every opportunity.*" (The Committee on the Restructuring of the Police Service 1984, p. 138)

Factors such as corruption, coercive tactics and quasi-militarisation were previously highlighted in studies by Deosaran (2002); Townsend (2009); King (2009) and Pino and Johnson (2011) as being instrumental towards policing in T&T, but they did not specifically explore the role and function of the police. Therefore, according to the author's knowledge, there is no previous empirical study which attempted to explore the role and function of the T&T police.

### 1.3. Aim for the Study

This study was aimed at exploring citizens perception of what they consider to be the key role and function of the police towards improving their lives and communities. To achieve this, the following research questions needed to be answered.

1. Does the present role and function of police officers in T&T satisfy the needs of citizens, for example, safety, security, problem solving, minimising the fear of crime, crime prevention and resolution?
2. Are the role and function of police officers aligned to a "force or service"?
3. What is citizens expectation of change towards the role and function of the police in T&T?

The aim of this study is not to form a generalisation of the findings and conclusion, but to develop a foundation of policing research on the topic within a T&T context on which further research can develop. The conclusions and recommendations can be used to enhance policing in T&T and other societies to develop a modern institution.

### 1.4. Justification for This Study

Previous research on public perception of the police focused on citizens demographic characteristics, such as race/ethnicity, socioeconomic status, age, and gender in developed countries, for example Australia, Canada, USA, and UK (Brown and Benedict 2002; O'Connor 2008). In T&T, there is a nascent body of literature that exists on citizen perceptions of the police, and at the time of this research, there was no specific study found,

which attempted to study citizens perception of the role and function of police officers. As a result, this study was aimed at using citizens (adults) views, opinions and expectations and testing it in a T&T context to generate an understanding of the topic with the intention to expand the present literature to include a post-colonial nation. Due to ethical principles, this study did not involve citizens under 18 years of age.

## 2. The Present Literature

### 2.1. Earlier Perceptions of the Role and Function of the Police

To understand the role and function of the police in society, it is imperative to examine the work of Banton (1964) "*the policeman in the community*" which suggested that police officers are primarily observers of the peace and not law enforcers. He argued that officers monitor community activities and respond to citizen calls for assistance. His work suggested that discretion was imperative to the role and function of the police as it permits officers to exercise moral judgement on when the law should be enforced because officers were persuaders rather than prosecutors. However, prior to Banton (1964), La Fave (1962) and Williams (1954) argued that the role and function of the police should not provide officers with discretion because it was an opportunity for bias and delinquency.

A study by Cain (1973) suggested that the role and function of the police was best defined as order maintenance and law enforcement. However, Bayley (1985) and Uglow (1988) argued that the police role and function are aligned to controlling some level of political authority and do not exclusively maintain law and order. According to Mawby (2003) the role and function of the police varies in accordance with systems, cultures, and country. He highlighted that the most basic role and function of police is law enforcement and order maintenance, and sometimes incorporating other activities and tasks. The suggestions made by Mawby (2003) were supported by Reiner (2010) who stated that in the UK, a quarter of police time was spent assisting people with mental health conditions and attending to other vulnerable groups. Reiner (2010) suggested that the role and function of the police in liberal societies have often become controversial on how best to develop a specific definition. For example, "a force" where officers primary function is law enforcement and order maintenance or "a service" whereby officers' primary function is to assist citizens and pacify social problems. Manning (2014) argued that the role and function of the police is dependent on a given society and its history, culture, and social construction. He suggested that policing a stable society differs significantly from a problematic society. Therefore, the role and function of the police is difficult to contextualise.

An integral aspect of the police role and function is to account for decisions made and the process used in making such decisions (Mac Vean and Neyroud 2012). Decisions are important towards the outcome of a situation and therefore, morals and ethics are paramount (Delattre 2011) and officers are often challenged between personal and professional ethical decisions. Consequently, before officers make any specific decision, they should rely on professional ethical standards because they are acting on behalf of the police institution and the population and not act on personal attributes (Delattre 2011).

### 2.2. Service Oriented Policing Model

Over the past decades, police institutions have been experiencing several events of reform efforts due to the traditional model of policing being impersonal, ineffective, and costly (Greene 2000; Hill 2020). Police institutions carried increased liabilities due to the use of force and lack of training which was being scrutinised by citizens and policymaker (Hill 2020). During this time there was the establishment of Community Oriented Policing (COP), Problem Oriented Policing (POP), Evidence Based Policing (EBP) and Body Worn Video cameras (BWV) which was all designed to reduce the use of lethal force (Hill 2020). However, these advancements were soon overwhelmed by a new shift in policing which became known as the Service Oriented Policing (SOP) model. This model of policing relied on the motto "to protect and serve" which promoted a foundation for transparency, approachability and accountability within police institutions and their officers (Gill et al.

2014; Hill 2020). This model of policing was focused on reducing crime and its fear, and community disorder whilst increasing citizens satisfaction and legitimacy with the police (Hill 2020).

The business of policing is traditionally law enforcement, with communities and citizens being the customers. Within recent times the police have been providing a plethora of services to communities, and these often vary in accordance with the demographic's construction of individual community (Hill 2020). In the United States, police services are usually classified into four categories. The first category is a service to each citizen by attending, assisting, and resolving their problems. The second is a service to violators. The police have a duty to "protect and serve" therefore, violators being citizens should be treated in a courteous and professional manner. The third is stakeholders such as courts, children services and mental health institutions to name some. It is important for police institutions to form a multiagency approach with these stakeholders to provide a more diverse range of support and service to citizens and communities. The fourth category is police officers providing a support service to each other. It is paramount that officers support and establish a well-being environment for each other since their well-being is likely to impact their performance and service to citizens and communities (Hill 2020).

A study on Service Oriented Policing by Scheider et al. (2003) found that service-based policing increased satisfaction within the role and function of the police. Another study by Pope and Pope (2012) suggested that SOP was responsible for enhancing citizens lives which in turn promoted safer communities, tranquil and developed environments and these all contribute to dramatic increases in property prices. According to Gill et al. (2014) SOP reduces citizens negative perception of crime in their communities and simultaneously increases their legitimacy with the police (Hill 2020). Whilst the SOP appeared to have been successful on many occasions, Hill (2020) argued that this model of policing did have some barriers. It was stated that SOP needed the participation of police officers, and this sometime could become contentious as some officers believe that their job was becoming that of a social worker. Hill (2020) argued that these officers did not believe that citizens were the customers of policing and thus, SOP was not a legitimate concept of policing. It was further established that budgets to police institutions often created financial implications on policing resources such as training and availability of officers (Hill 2020).

*2.3. Colonial Policing Model*

The Colonial Policing model was based on the ethos of the Royal Irish Constabulary (RIC) which was designed to suppress political disorders in Ireland (Sinclair 2006; Mathura 2019). The RIC mirrored the Imperial Army since many officers were ex-soldiers and maintained a military persona such as foot drills, firearm training, and public order duties (Tobias 1977; Anderson and Killingray 1991). When the British Empire began to expand, this model of policing was transferred to new colonies, but was often met with hostility and rejection from local citizens due to cultural differences. According to Anderson and Killingray (1991), all senior officers (gazette) and most inspectors were Caucasians (white) and were recruited from the army because of their military training and persona (Tobias 1977; Cole 2003). Mawby (2003) suggested that the recruitment of white senior officers was solely to create a divide between the police and local citizens whilst Junior officers (constable to sergeant) were recruited locally or from other colonies (Sinclair 2006; Bell 2013). This ensured that indigenous officers never attained a management rank which could have jeopardised the objectives of colonisation (Tobias 1977; Sinclair 2006). Stanislas (2014) stated that most colonial officers were recruited with poor educational attainment and Sinclair (2006) highlighted that it was important for officers of the colonies to favour sports, be no more than 35 years and be physically fit and well-built. Bell (2013) suggested that colonial officers were responsible for the colony's protection from internal and external threats, ensuring the local workforce were law compliant and protecting the foreign traders. Arnold (1986) highlighted that colonial police officers was not restricted to law enforcement duties but often performed the role of judge and jury by providing rapid punishment onto local

citizens. According to Mars (1998) policing on colonies often relied on coercive and violence tactics onto local citizens (Jefferies 1952; Brewer 1994). Arnold (1986) argued that officers used coercion to prevent local citizens from challenging the colonial power and authority (Das and Verma 1998; Bell 2013). As a result, colonial officers developed a reputation for being rude, unhelpful, violent, and unsympathetic with the local people causing a lack of cooperation and strained relationships (Anderson and Killingray 1991; Cole 2003). In the colonies, officer's behaviours frequently became a concern; for example, corruption, viewed by citizens as enemies and oppressors, using intimidation tactics, agents of the state and coercion which caused the local communities and citizens to distant themselves from the police (Arnold 1986; King 2009; Bell 2013). The colonial policing system became known as a state's apparatus of authority and to fulfil the aims and objectives of the colonial government, whilst ensuring that the local citizens (subjects) were law compliant and did not resist (Deosaran 2002; King 2009; Mathura 2019).

### 2.4. Trinidad and Tobago Policing Service

The colonial model of policing was introduced to T&T when the British ceded the country in 1797 (King 2009; Wallace 2011). The senior officers were all from England and Ireland with military backgrounds and junior officers were recruited locally or brought from other colonies (Johnson 1991; Sinclair 2006). By 1843, the T&T police force had 12 police stations and approximately 100 officers across the country (De Verteuil 1986; Pino 2009). Prior to the country's independence in 1962, policing duties were mainly focused on political tensions and state affairs (Anderson and Killingray 1991; Brereton 1996). By the late 1960s the police force was officially recognised as a service and adopted the name, Trinidad and Tobago Police Service (TTPS).

There have been several efforts to reform T&T's police institution which dates back to 1959 with more than 200 recommendations for improvements (Job 2004; Mathura 2019). The Lee committee in 1959 recommended changes to the rank structure, in 1964 the Derby committee recommended administration upgrades, accountability procedures, higher education and training and advanced investigation techniques. The Carr committee in 1972 recommended changes for effectiveness and efficiency, in 1984 the Bruce committee recommended a comprehensive restructure of the TTPS and the O'Dowd (1991) committee recommended improved resource management, advanced training and revised duties for all officer (Job 2004; Mathura 2019). However, the majority of these recommendations were ignored in government (Job 2004; Mathura 2019). Apart from reform committees, a nascent body of policing research in T&T have recommended various institutional changes. Authors such as Johnson et al. (2008), King (2009), Wallace (2011), Pino and Johnson (2011), Ryan et al. (2013), Seepersad (2016) and Adams (2019) have recommended several policy and practice changes within the TTPS. However, many of these recommendations were ignored by the governments and police executives (Job 2004; Maguire et al. 2017; Watson and Kerrigan 2018).

A community policing study by Deosaran (2002) highlighted that citizens in T&T were dissatisfied with the TTPS because of some officer's delinquent behaviour such as the use of brutal and excessive force towards citizens, accepting money from criminals to destroy evidence, renting of service firearms to criminals and giving special treatment to friends and family. Another study by Wells and Katz (2008) showed officers inabilities to prevent and solve crimes. A study by Townsend (2009) highlighted that some officers were instrumental in the illegal drug trade (Scott 1984; Griffith 2000) and gang activities (Pawelz 2020). The Police Complaints Authority of Trinidad and Tobago (PCA) reported that since 2005 they have been receiving high volumes of complaints from citizens annually, and during 2014 and 2018, they had received a consistent increase in the number of complaints made against police officers (Police Complaints Authority of Trinidad and Tobago 2003–2019; Mathura 2019).

Some experiments with foreign police strategies were tried in T&T due to the escalating crime rates. However, they mainly adopted an American and European "one size

fit all" approach (Pino 2009; Watson 2016; Watson and Kerrigan 2018). These ideological assumptions and approaches failed because North American and European societal dysfunctions differed from T&T. These strategies did not acknowledge the differences and implications between the different societies, cultures and crime patterns (Harkness et al. 2015; Watson 2016; Watson and Kerrigan 2018). According to Watson and Kerrigan (2018), policing and crime fighting strategies in T&T has been historically influenced by political affiliation, poor police management and neglect for citizens involvement (Maguire et al. 2017; Watson and Kerrigan 2018). These factors collectively reduced community support for police initiatives and reform efforts remained challenged (Maguire et al. 2017; Watson and Kerrigan 2018; Mathura 2019).

A major problem with the TTPS was the inequality of male to female ratio (Job 2004; Mathura 2019). The institution continued to be influenced by its colonial heritage of being male dominated because men were perceived to be stronger than women (Sinclair 2006). Women in the TTPS were mainly bound to administration duties because of this perception and the recruitment of female officers have been much lower than that of their male counterparts (Deosaran 2002).

A fundamental problem that previous studies and reports on the TTPS failed to address was a more precise definition towards the role and function of police officers. Whilst the deficiencies of the institution and officers were imperative for reform efforts, a more precise account for the role and function is paramount as this can guide reform efforts such as recruitment, training and the needs of the communities and citizens. As a result of the "gap" in the current literature, this study is aimed at addressing the deficiency.

## 3. Methodology

### 3.1. Research Design

Considering the lack of research on policing in T&T, this research design was aimed at exploring citizens' perception of the police role and function in a post-colonial nation, namely Trinidad and Tobago. This study used a qualitative design which provided descriptive information from participants. This type of data was best obtained from one-to-one in-depth interviews where participants were given an opportunity to fully express their views and opinions. According to Bryman et al. (2022) qualitative data provides a better understanding and evaluation of the various aspects and social dimensions of people's life, attitude, and experiences.

### 3.2. Research Population and Sample

Trinidad and Tobago are a twin-island nation; the most Southern Caribbean islands and North-East of the South American continent. The islands have a combined population of approximately 1.3 million people from diverse backgrounds, but the two major groups are from African and Indian ethnicities (Brereton 1996; Mathura 2019).

The general population of T&T were more likely to have individual perceptions about the TTPS, officer's role and function in the communities, and citizens expectation from the institution. This study consisted of 45 participants who were members of the public and represented the various geographical locations of T&T, for example, Point Fortin and San Fernando in the South; Mayaro and Sangre Grande in the East; Caroni and Chaguanas in the Central; Port of Spain and Curepe in the North, Maraval and Chaguaramas in the West and Crown Point and Charlotteville in Tobago. The selected areas consisted of rural and urban communities which provided a balanced representation to the data collected. Participants were also selected on demographic characteristics (see Table 1). The interviews were conducted using qualitative open-ended questions and recorded using a Dictaphone.

**Table 1.** Demographics of Participants.

| Sexual Orientation | Quantity |
| --- | --- |
| Male | 20 |
| Female | 21 |
| LGBT+ | 3 |
| Other | 1 |
| **Race/Ethnicity** | **Quantity** |
| Afrotrinidadian | 13 |
| Indotrinidadian | 13 |
| White | 6 |
| Mixed | 13 |
| Other | 0 |
| **Age Group** | **Quantity** |
| 18–30 | 16 |
| 31–50 | 16 |
| 50+ | 13 |
| **Income** | **Quantity** |
| Unemployed | 9 |
| Employed (full-time) | 20 |
| Self Employed | 9 |
| Retired (pension) | 5 |
| Student | 2 |
| **Marital Status** | **Quantity** |
| Single | 20 |
| Married/In a Relationship | 20 |
| Widowed | 10 |
| **Education** | **Quantity** |
| No Formal | 9 |
| Primary | 13 |
| Secondary | 13 |
| Tertiary/University | 10 |

The snowball sampling technique was used to recruit participants for this study because the author was not T&T based. Snowball sampling provided a link between citizens who had an interest in the study, generated a large pool which final participants was chosen from and promoted diversity (Parker et al. 2019; Bryman et al. 2022). Participants were not offer financial incentives, and was told that their participation was voluntary, and could discontinue at any time. No personal information was recorded, and all data would be destroyed after analysis and publication. For ethical reasons, citizens under the age of 18 years were not recruited for this study.

*3.3. Data Analysis Method*

This study used the thematic analysis approach from Braun and Clarke (2006). According to Maguire and Delahunt (2017), Thematic Analysis (TA) is a useful analytical approach for finding patterns and themes in qualitative data. Braun and Clarke (2006) proposed a six-stage analysis in TA which, when becoming familiar with the data, generate initial codes, begin to search for themes, review the themes, define the themes, and write up. For this study, TA was considered most applicable because of the lack of study on policing in T&T and furthermore on citizens perception on the role and function of the police in a post-colonial nation. As a result, TA allowed themes to be extracted from the data collected which produced variables for further exploration and to construct a foundation for future studies.

## 4. Findings

Research Question 1: Does the present role and function of police officers in T&T satisfy the needs of citizens for example, safety, security, problem solving, minimising the fear of crime, crime prevention and resolution?

Most participants (*n* 41) stated that police officers in T&T had a distant and fragile relationship with the citizens and communities. Participant explained that many police officers were lazy and unhelpful towards citizens. It was highlighted that T&T have been experiencing an increase in violent crimes especially with the use of firearms. However, the police were rather reactive to these problems and on some occasions, officers were renting their service firearms to the criminals. Participants indicated that they did not feel safe in their communities especially at night and felt fearful of criminals. It was stated that because of the poor citizen and police relationship and officer's poor attitude towards their job, the TTPS was not able to prevent and solve crimes in T&T. Most participants had first-hand experience and others had acquaintances who approached the police for assistance and were often told that there was a lack of resources and as such assistance was not guaranteed. Participants further stated that if the police responded, officers were often rude and aggressive towards victims which often developed confrontational situations and coercive tactics being used by officers.

According to these participants, most police officers in T&T did not have the skills and knowledge on how to resolve problems and provide advice. However, male officers were found to be more interested in sexual relationships with females in the communities, but female officers were identified as being more helpful and sympathetic. Participants were of the view that officers should receive training on problem solving, and communication techniques which would be applicable to citizens and communities. A 22-year-old female participant stated:

> *"These police officer in T&T only want to harass women for sex. If you have a husband or man, they try to come to your house when he not at home. But violent crime is out of control in T&T, and they cannot fix that. We have teenagers running around with guns and shooting innocent people and the police have no idea who they are. Most people do not want to speak with the police because they do not care and some of them even involved with the criminals. The women officer a bit better because they will listen and try to help but you do not see them often."*

The minority group of participants (*n* 4) stated that they had a good relationship with the police and found officers to be helpful based on prior experience or those of acquaintances. These participants explained that there were some violent crimes in T&T and citizens needed to consider their actions and not accuse the police of being ineffective. When asked about police corruption, participants highlighted that there were a small number of police officers involved in delinquent behaviour and all should not be labelled equally. These participants stated that their [and acquaintances] encounters and experiences with the police were positive, and they feel safe in their communities because the police conduct regular welfare checks and patrols. It was further highlighted by these participants that if they had a problem, they felt confident to approach the police for assistance and have their problem resolved. A 57-year-old male stated:

> *"Some of these T&T people not easy, they don't want to work hard and live an honest life. They always stealing, robbing, selling drugs, prostituting themselves and killing innocent people. When the police arrest them, they quick to saying the police wicked and not helping them."*

Participants Demographics: The participants from the majority group were from disadvantaged and middle-class communities located on the outskirts of urban areas and some rural areas to a smaller extent. These participants were between the ages of 18 and 50 and were of Afrotrinidadian and mixed ethnicity with some Indotrinidadians to a lesser extent. There was a balance of males and females, employed, self-employed and a small number of unemployed. Most held a primary or secondary school education. The minority

group of participants were from affluent communities, were above the age of 40 years, held a university qualification and were employed with a small number being self-employed business owners. They were of White and Indotrinidadian ethnicity, a balance of males and females and were all married.

Research Question 2: Are the role and function of police officers aligned to a "force or service"?

The majority of participant (*n 39*) were of the view that the TTPS and its officers were comparable to a "force". These participants highlighted that the institution (including officers) minimally provided a service to the citizens and communities. It was further explained that the priority of the police in T&T appeared to be law enforcement where officers were focused on arresting minor offenders. These participants explained that the word minimal was used because the institution did provide services such as liquor licenses, firearms licenses, and criminal record checks. However, police officers seldomly patrolled their communities and offer guidance, support, and advice to citizens. Additionally, these participants believed that many officers were aggressive and forceful in their demeanour and attitude towards citizens. It was stated that citizens sometimes have problems and needed assistance. However, officers were not able to assist. A 77-year-old male stated:

> *"I am from the old colonial day when Trinidad and Tobago were a British colony. The policemen back in those times was wicked and heartless. The used to beat people up for no reason and the women officers did not have a voice because the male officers were in control. Those officers did not care about helping people, they cared about themselves. These young police officers now have a bit more education but they just driving around in fancy cars and vans but not really helping people. People often have problems and not sure who to talk to or how to resolve it, but these police nowadays have no time for helping people or resolving problems, therefore crime is out of control in this small nation. These officers just want to make a quick arrest to look good and say they are working hard."*

The minority group of participants (*n 6*) stated that they viewed the TTPS and its officers to be a "service". These participants explained that their contact with the police were positive, officers were able to resolve their problems and was always professional. They stated that the police provided a service of security and safety for their communities by conducting regular patrols, were friendly with citizens and would assist when required. A 44-year-old female stated:

> *"I never had a problem with the police. They are always friendly and professional in doing their job. If I had a problem and called them, I know they will help to resolve it."*

Participants Demographics: participants from the majority group were from disadvantaged and middle-class communities which were located on the outskirts of urban areas with a small number from rural communities and were between the ages of 18 and 79 years. There was a balance of ethnicities [Afrotrinidadian, Indotrinidadian and mixed], males and females, employed, self-employed and unemployed. Most held a primary or secondary school education with a small number having a university qualification. The minority group of participants were from affluent and middle-class communities, between the ages of 35 and 55 years old, held university qualifications and were employed with a small number self-employed. They were of White and Indotrinidadian ethnicity, a balanced of males and females and most were married with a small number being single.

Research Question 3: What is citizens expectation of change towards the role and function of the police in T&T?

The majority of participants (*n 40*) stated that the TTPS and its officers needed a modern approach to policing. It was indicated that citizens often experience modern problems [financial and social] and officers were not able to assist. Participants explained that in recent times after the COVID-19 pandemic, the T&T economy became strained, and citizens found life difficult due to unemployment and poor mental health. However, police officers were not trained to offer advice and simultaneously navigate citizens (especially the younger ones) away from crime. A further problem identified by these participants was

technology (social media) which had created a platform for delinquency amongst younger citizens. However, most police officers were not aware of the criminal activities on these platforms and how to effectively address them. These participants highlighted that many young people who were from the criminal fraternity often used social media platforms to communicate via smartphones. However, most police officers were not familiar with this technology and were helpless in detecting such crimes and keeping the communities safe.

According to these participants, it would be beneficial to have officers and citizens work together via technological platforms to share information more effectively and efficiently whereby the police could be notified of important information. Participants also highlighted that police officer training needed a more contemporary approach, in accordance with the problems that citizens and communities experience. It was stated that different communities in T&T experience different problems, so departmental training and continuous professional development was paramount towards serving the citizens of T&T. Participants further stated that officer's attitude towards citizens was a major concern for change. They explained that officers should have a more understanding and helpful approach towards citizens and less of a rude and aggressive attitude. A 27-year-old male stated:

*"In T&T we need a more professional police service, too much of these officers have that old mentality. Times has changed, people have changed, and the police need to change also. Long ago police did what they want, when they want, and no questions asked. But now they need to be accountable for their actions. They need better training, more advanced skills to help people, not apply force on people. I would personally say they need communication, problems solving and some social welfare training. Another big issue is the young generation now is into technology and with a smart phone they are doing everything, but most officers don't know how to use a smartphone. So, these young criminals make officers look old fashion and outdated, the youths ahead of the police."*

The minority group of participants (*n* 5) stated that they viewed police in T&T as professional, had good knowledge, skills and were helpful. These participants highlighted that their experiences with the police were positive, and the officers did everything possible to assist them. As a result, they did not feel that the TTPS need any immediate change. However, these participants acknowledged that there have been some media coverage of minor cases of unprofessional behaviour and activities by a small number of officers. Therefore, some future improvements to accountability could become useful. A 33-year-old male stated:

*"As far as I am concerned the police do a good job and have the right skills to do it. I don't think they need any immediate change but as times and things change, they might need to keep updated. However, that's not an emergency or something for immediate actions."*

Participants Demographics: participants from the majority group were from disadvantaged and middle-class communities located on the outskirts of urban areas, a small number were from rural communities and a small number from urban areas. These participants were between the ages of 18 and 75 years old. There was a balance of ethnicities [Afrotrinidadian, Indotrinidadian and mixed], males and females, employed, self-employed and unemployed. Most held a primary or secondary school education with a small amount having a university qualification. The minority group of participants were from affluent and middle-class communities, between the ages of 35 and 60 years old, held university qualifications and were employed with a small number self-employed [business owners]. They were of White and Indotrinidadian ethnicity, a balanced of males and females and most were married with a small number being single.

## 5. Discussion

The police in society are a representation of the state's apparatus for social and moral compliance. Police institutions were established to prevent crimes, the associated fear of crime, victimisation and to allow community safety whilst simultaneously promoting

the development of sustainable communities (Mathura 2019; Sani et al. 2022). Whilst the police role and function are important to all societies, it was imperative to gain a better understanding of a post-colonial context such as T&T to determine the needs of citizens and institutional changes in a country that has been experiencing constant modernisation. Understanding citizens' perception of the role and function of police in a specific setting is more likely to inform and educate practitioners of the problems and needs of these citizens, but equally important is the services that the police institution can develop to serve the communities. Briefly, Service Oriented Policing (SOP) seek to provide a service to citizens, resolve personal and community problems, increase citizens satisfaction with the police, enrich the quality of life for citizens and maintain the moral fabric of communities (Scheider et al. 2003; Pope and Pope 2012; Hill 2020). Citizen's satisfaction with the police is more likely to develop trust and confidence between citizens and the police, promote information sharing on safety concerns and demonstrate that the police care about citizens and community problems (Merenda et al. 2021; Sani et al. 2022).

In this study, most participants described police officers as being lazy, rude, unhelpful, corrupt, and unskilled. These factors highlighted by participants were important towards understanding the need for future development of the TTPS and parallel with participants having negative and less favourable perception of the police and their role and function in the community. According to Bolger et al. (2021) positive perceptions of the police is important towards compliance of the law and citizens networking with the police to foster safer communities (Sani et al. 2022). The finding of this study appears to be similar to that of studies by Deosaran (2002) and King (2009) who suggested that the TTPS needed to become a modern institution to serve the needs of the citizens and minimise the use of coercive tactics by officers. These studies also made recommendations to improve police accountability, allowing officers to take responsibility for what they did and minimise any form of malpractice that might affect TTPS relationship with citizens. Whilst several studies have previously identified the need for a modern TTPS, there were minimal recommendations for it to be transformed into a service-oriented institution.

A notable observation associated with this study was the difference in participants responses and their socioeconomic status. Participants who formed the majority group represented the disadvantaged and middle-class communities to a higher extent and the affluent communities to a lesser extent. On the other hand, participants who formed the minority group represented the affluent communities to a larger extent and the middle-class to a lesser extent with no representation from the disadvantaged communities. As a result of these findings, it could be suggested that in T&T, the participants from disadvantaged and middle-class communities held negative or less favourable perceptions of the police role and function, were less satisfied with the help and assistance from officers and were less likely to approach the police. Based on the findings of this study, it could be further suggested that participants from the affluent communities and middle-class (to a lesser extent) held positive or favourable perceptions of the police and their roles and functions. As a result, these participants were more likely to be satisfied and approach officers for assistance. These findings are similar to studies conducted by Brown and Benedict (2002) and Mathura (2019) who suggested that the citizens' perceptions of the police often vary in accordance with their status on the socioeconomic hierarchy.

Participants demographic variables were instrumental towards obtaining a spectrum of responses that would promote inclusiveness and robustness to this study. According to the findings, there was a balance between those identified as male and female, with a minimal number identified as other sexualities. Ethnicities varied in accordance with groups characteristics. The majority group was comprised of predominantly Afrotrinidadian, Indotrinidadian and Mixed-race participants whilst the minority group comprised of predominantly Whites and few Indotrinidadians. There was a good balance between married and single participants and employed, self-employed with a minimal number of unemployed participants. However, the majority group of participants predominantly held a secondary school education and lower with only a small number with a degree

qualification. The minority group on the other hand predominantly had a degree qualification and a small number stated they were self-employed (business owners). Throughout this study there was a good variety of age range amongst the participants. However, the environment variable fluctuated since most of the participants from the majority group were from semi-urban communities and a smaller amount from rural areas. The minority group of participants were all from semi-urban communities, but these communities were private housing complexes. According to Webb and Marshall (1995) citizens demographic characteristics play an important role towards how they perceived the police, and this is because people are different, and their needs and problems vary. The demographic variable found in this study is aligned to those found in a study by Webb and Marshall (1995) and was later echoed by Brown and Benedict (2002).

## 6. Conclusions

### 6.1. Summary of Discussion

The aim of this study was to explore citizens' perception of the role and function of the police in a post-colonial nation. To achieve this, three research questions were used.

1. Does the present role and function of police officers in T&T satisfy the needs of citizens for example, safety, security, problem solving, minimising the fear of crime, crime prevention and resolution?
2. Are the role and function of police officers aligned to a "force or service"?
3. What is citizens expectation of change towards the role and function of the police in T&T?

The findings from this study suggested that the TTPS needed to implement a Service Oriented style of Policing (SOP) capable of preventing crime, the fear of crime, victimisation and simultaneously promoting citizen and community development. If implemented, this will foster trust and confidence in the police, better communication, relationships between citizens and the police, and officers will be able to serve the communities better by providing guidance and support to citizens.

According to the findings from this study, there is a need for revised training of police officers in T&T. It could be suggested that advanced training, which is focused on officers being understanding and helpful, able to resolve problems in a professional manner, develop skills on social welfare and community relations would be able to serve the needs of citizens. It could be suggested that officers in the TTPS be given Continuous Professional Development (CPD) in accordance with the problems being experienced in the specific communities they serve, since problems often vary according to environmental context.

It was also identified that there was the need for improved accountability in the TTPS so that officers can be held accountable for their actions and omissions. The findings from this study suggested that there was police delinquency in the TTPS, and officers were not being held accountable. As a result, improved accountability would facilitate officers taking responsibility for their actions, become less reliant on the use of coercive tactics and use their persuasion skills and abilities, stop sexual harassment towards females in T&T and approach citizens in a professional manner.

### 6.2. Theoretical Implications

When studying citizens' perception of the police role and function in any society it is important to consider demographic variables such as age, socioeconomic status, community, race/ethnicity, gender, and education. These variables showed significant importance and fluctuation throughout the findings of this study and could be applicable to other societies. It is also imperative to conduct research on citizens perception of the police role and function using environmental context (urban and rural), since these variables played an important role in the findings of this study. Additionally, an important aspect of this study was the model of policing use and the type of training associated with the model. When researching citizens perception of the police role and function, it is important to examine the model of policing and style of training to gain a better understanding.

### 6.3. Future Research

The findings of this study highlighted that environmental context (urban and rural) was crucial towards participant perception of the police role and function. However, there is no previous research (at the time of this study) that attempted to examine this variable independently. As a result, it would be productive to conduct future research on this variable. This study used a qualitative method of 45 participants which could not be used towards generalisation. Therefore, it would be beneficial to conduct a quantitative study to obtain a great volume of responses which could be used for generalisation.

**Funding:** This research received no external funding.

**Institutional Review Board Statement:** Not applicable.

**Informed Consent Statement:** Informed consent was obtained from all subjects involved in the study.

**Data Availability Statement:** Not applicable.

**Acknowledgments:** Thank you to the participants for their time and contributions, peer reviewers for their constructive feedback and editors. Thank you to Jan Ludvigsen from Liverpool John Moores University for his guidance.

**Conflicts of Interest:** The authors declare no conflict of interest.

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
