# Peer review of "Exploring Citizens Perception of the Police Role and Function in a Post-Colonial Nation"

_socsci, doi:10.3390/socsci11100465_

Round 1
Reviewer 1 Report
This manuscript seeks to improve police-community relations in the country of Trinidad and Tobago (Trinidad). It notes that previous scholarship has considered many aspects of policing in Trinidad, but not overall model of policing employed there (its "role and function"). It uses 45 qualitative interviews to clarify citizen perceptions of the policing role and function. The author(s) make a recommendation for a Citizen-Oriented and Problem Solving (COPS) model based on their analysis of previous scholarship and their interviews.
The interview data are barely employed or discussed in any depth. There is only one quotation from this data. Although a couple of references are made to "the majority of participants," the manuscript does not give a full sense of what the data show. What questions were participants asked? How many were male or female? Where all of the interviews conducted on the more populous island of Trinidad (perhaps in the major city of Port of Spain), or were they distributed throughout the country? Where they dissenting voices from the majority? Both the methodology and findings sections should be much more detailed.
Finally, the COPS model suggested in the conclusion appears out of nowhere. If it is to be the final recommendation, then it should be discussed earlier on in the literature review. That said, it is unclear to me whether the adoption of a COPS model will address most of the concerns about Trinidadian policing that come up in the article: corruption, bribes, participation of police in illegal activities, and the use of physical force to maintain inequality.
Author Response
Please find an attachment with how I have taken your feedback and implemented it into my manuscript. Thank You Warm Regards

Reviewer 2 Report
Introducing a community policing strategy in post-colonial societies presents a special challenge for the police organization. It is a complex and lengthy process. It implies a comprehensive approach to structural, organizational, tactical, personnel issues in the police organization itself. At the level of society and the legislative framework, it should be a framework for the protection of human rights.
Therefore, the research of POLICE in post-colonial societies should represent a significant contribution to the analysis of positive and negative experiences of this strategy of police work in the specific context of the post-colonial one.
As much as an original methodical approach and research contribution was expected from this research, the presented work did not adequately respond to the set research problem.
I noticed the following disadvantages:
- The frame of reference is outdated and outdated. The average "age" of used references amounts to more than 10 years. This observation is particularly related to the experiences of the police in the application of COPS. At the same time, the essence of the concept itself is not adequately and experientially explained.
- The methodical approach, the essence and nature of the research itself, the structure of the sample, the research questions, the method of realization of the research, its spatial and temporal framework are not clearly presented and explained.
- In presenting the findings of the research, the responses of the participants/informants should have been anonymously stated, and recorded in full (as on page 5). Therefore, the quotes of the participants' answers to the interview questions, their individual opinions, are missing. The answers are generalized, and we don't even know how the questions were worded in the interview.
- The introductory part of the Discussion methodologically belongs to the theoretical aspects of the work or the literature review.
- The conclusion is relativized. The author concludes without arguments, without using the findings of his own research. This approach is particularly represented in the analysis of COPS.
- In the paper itself, the author does not deal with the analysis of the function, application and importance of COPS in a specific context such as rural areas.
- The research does not contain an analysis of the gender dimension of the police organization in the researched society, nor does it contain a gender approach in the implementation of the COPS.
I believe that this research work requires numerous and complex corrections and changes, both in the methodical approach and in the conceptual level (theoretical basis of the research questions).
Author Response
Please find a document with the feedback provided and how I addressed these. Thank You Warm Regards

Round 2
Reviewer 1 Report
The revisions to this manuscript are excellent. It now reads as an important contribution not only to the scholarship on policing Trinidad and Tobago, but will be useful to those seeking to understand the role of policing and policing reform in any postcolonial nation. I hope to teach it in my classes following its publication.
Reviewer 2 Report
The author/s took into account the reviewer's recommendations and significantly improved the quality of his manuscript.
In this, new, corrected form, the manuscript meets the necessary conditions (formal and material) to be published in the Journal of Social Sciences.